# Body Mass Index and the Risk of Adult-Onset Asthma: A Prospective Observational Study among 59,668 Middle-Aged Men and Women in Finland

**DOI:** 10.3390/nu16152515

**Published:** 2024-08-01

**Authors:** Ville A. Vartiainen, Pekka Jousilahti, Jaakko Tuomilehto, Tiina Laatikainen, Erkki Vartiainen

**Affiliations:** 1Heart and Lung Center, Helsinki University Hospital, 00290 Helsinki, Finland; 2Population Health Unit, Finnish Institute for Health and Welfare, 00271 Helsinki, Finland; pekka.jousilahti@thl.fi (P.J.); jaakko.tuomilehto@thl.fi (J.T.); tiina.laatikainen@thl.fi (T.L.); erkki.vartiainen@thl.fi (E.V.); 3Department of Public Health, University of Helsinki, 00014 Helsinki, Finland; 4Saudi Diabetes Research Group, King Abdulaziz University, Jeddah 21589, Saudi Arabia; 5Institute of Public Health and Clinical Nutrition, Faculty of Health Sciences, University of Eastern Finland, 70211 Kuopio, Finland

**Keywords:** asthma, body mass index, weight, height, incidence, obesity

## Abstract

**Introduction:** Obesity, in addition to many other negative health consequences, affects pulmonary function and is a potential risk factor for asthma. **Methods:** We analyzed the association of body mass index (BMI) with incident asthma among 60,639 Finnish men and women aged 25 to 74 years who participated in a population-based chronic disease risk factor survey in 1972, 1977, 1982, 1987, 1992, 1997, 2002, 2007, or 2012. Data on lifestyle factors such as smoking and physical activity, as well as medical history, were obtained, and various physical measurements, including height and weight, were taken at baseline. Incident asthma events were ascertained from the National Social Insurance Institution’s register data. The study cohorts were followed-up until the end of 2017 through registers. **Results:** During the follow-up, 4612 (14%) women and 2578 (9.3%) men developed asthma. The risk of asthma was analyzed in the following three BMI categories: <24.9 (reference category), 25–29.9 (overweight) and ≥30 kg/m^2^ (obesity). Hazard ratios (95% CI) were 1.34 (1.24–1.43) and 1.57 (1.44–1.71) in women and 1.25 (1.14–1.37) and 1.63 (1.44–1.83) in men. The observed association was independent of smoking, height and leisure-time physical activity. In women, 30.8% (19.2% in men) of the total asthma incidence was attributed to overweight and obesity. **Conclusions:** Overweight and obesity are important risk factors for asthma.

## 1. Introduction

Obesity is a major public health problem in the world [1,2]. It is associated with an increased risk of many diseases and health hazards, such as cardiovascular and cerebrovascular diseases, hypertension, diabetes, and musculoskeletal disorders [3,4,5]. It has also been shown that obesity and weight changes affect pulmonary function [6,7,8]. Both in high-income and middle- and low-income countries, the prevalence of obesity and its negative health consequences are rapidly increasing [1,9]. Adult-onset asthma is defined as asthma presenting for the first time in adult life. It tends to be non-allergic and often does not respond to inhaled corticosteroids as well as allergic asthma does. Similarly, asthma with obesity may present with different patterns of airway inflammation but typically with limited eosinophilic inflammation [10].

The US Nurses’ Health Study II was the first longitudinal study demonstrating a temporal association between obesity and the risk of asthma [11]. The Canadian National Population Health Survey also found a significant direct association between body mass index (BMI) and the risk of asthma among women but not among men [12]. In both studies, the length of follow-up was relatively short, corresponding to four and two years, respectively. More recently, a meta-analysis comprising of seven studies including 333,102 people found increasing odds of high BMI with incident asthma in a dose-dependent manner [13].

The aim of the present study was to prospectively analyze the association of overweight and obesity with adult-onset asthma incidence among a large population-based cohort of middle-aged men and women. In addition, we determined whether the association is affected by age, smoking, height or leisure-time physical activity and whether the length of follow-up modified the association.

## 2. Materials and Methods

### 2.1. Study Population

Methods of data collection have been described in detail elsewhere [14]. Nine cross-sectional risk factor surveys (National FINRISK study 1972–2012) were conducted in Finland every 5 years by the Finnish Institute for Health and Welfare (previously the National Public Health Institute). Independent population surveys were conducted in following five areas: in the provinces of North Karelia and Kuopio since 1972, in southwest Finland since 1983, in the Helsinki capital area since 1992, and in Oulu province since 1997. Random samples were drawn from the population register. The sampling was stratified according to sex and 10-year age groups and was modified to be compliant with the protocol of the World Health Organization’s MONICA project and recommendations of the European Health Risk Monitoring project upon their introduction. Participation gradually declined from 91% in 1972 to 63% in 2012. The current study includes 32,943 women and 27,696 men aged 25–74 years. People with asthma, chronic bronchitis, and chronic obstructive pulmonary disease at baseline were excluded from the analyses.

The study was conducted according to Finnish legislation at the time of each data collection, ethical rules of the National Public Health Institute, and the Declaration of Helsinki. Approvals from independent ethical committees that have varied over time were obtained for each individual study. In the first studies, informed consent was given verbally, and since 1997, written informed consent has been received.

### 2.2. Baseline Measurements

A self-administered questionnaire was mailed to the participants in advance. It included questions about smoking, physical activity and medical history. Using a set of standardized questions, the participants were classified into the following three smoking categories: current smokers, ex-smokers, and lifelong non-smokers (never smokers). In prospective analyses, ex-smokers who had not smoked for the past six months or more prior to the survey were considered non-smokers, and ex-smokers who had quit within the past six months were considered smokers. Leisure-time physical activity was assessed using the questionnaire, and the participants were classified into the following three categories: sedentary, moderately active and active. Medical history included questions about diagnosis with asthma or chronic bronchitis. Education was elicited in the questionnaire as full years of education, and educational tertile was then calculated. Educational tertiles were calculated separately for each 5-year birth cohort because of changes in the school system over the decades. At the study site, specially trained research nurses checked that the questionnaire was fully completed, and they measured height and weight using a standardized protocol. Height was measured without shoes. Weight was measured with light clothing. BMI (weight divided by height squared, kg/m^2^) was used as a measure of relative body weight. All measurements were conducted only at baseline.

### 2.3. Prospective Follow-Up

The study cohorts were followed until the end of 2017 through computerized register linkage using the national personal identification number assigned to every citizen in Finland. Data were obtained from three nation-wide registers, namely the Death Register, the Hospital Discharge Register and the National Social Insurance Institution’s Drug Reimbursement Register. The eighth revision of the international Classification of Diseases, injuries and Causes of Death (ICD) was used from 1969 to 1986, the ninth revision from 1987 to 1995, and the 10th revision was adopted at the beginning of 1996. ICD-9 code 493 and ICD-10 codes J45 and J46 were considered as asthma, and ICD-9 codes 491–492 and ICD-10 codes J41-J44 were considered chronic obstructive pulmonary disease.

Data on the occurrence of bronchial asthma were obtained from the National Social Insurance Institution’s Drug Reimbursement Register on persons entitled to special reimbursement for asthma drug costs. To receive such a special reimbursement, the diagnosis of asthma is usually assigned by a general practitioner or specialist in pulmonary medicine, allergology, internal medicine, or pediatrics, and the diagnosis needs to be based on objective findings in pulmonary function tests. The statements documenting these findings are then reviewed and approved by a consultant physician from the Social Insurance Institution. In selective situations, patients with particularly severe chronic obstructive pulmonary disease can also receive special reimbursement.

Persons who reported having asthma at baseline, who had prior history of a hospital discharge because of asthma, or who were entitled to special reimbursement for asthma drug cost before the time of baseline measurements were excluded from the prospective follow-up. We also excluded people who had been diagnosed with chronic obstructive pulmonary disease or chronic bronchitis either at the time of the baseline survey or during the follow-up. Thus, the study endpoint was newly diagnosed asthma based on the data of the National Social Insurance Institution’s Drug Reimbursement Register. All participants in the study cohorts were followed until either the defined endpoint, death or the end of the follow-up period. During the follow-up, by the end of 2017, 4612 women and 2578 men had been diagnosed with incident asthma. The numbers of participants and those excluded are reported in Figure 1.

### 2.4. Statistics

Standard t-tests and chi-square tests were used to compare the mean levels of continuous variables and the prevalence of categorical variables between subjects with and without incident asthma. In prospective follow-up, the association between BMI and asthma was analyzed using Cox proportional hazard models. The participants were classified in the following three BMI categories: <24.9 (reference group), 25–29.9 (overweight) and >30 kg/m^2^ (obese). The categories were included in the models as dummy variables. All models were adjusted for age, area and study year. Other variables included in the models were smoking as a categorial variable (dummy: lifetime non-smokers versus ex-smokers and current smokers) and educational tertile and leisure-time physical exercise as continuous variables. The association between BMI and the risk of asthma was assessed during the first five-year period after the baseline survey; more than 5, 10, 20, and 30 years after the baseline; and during the maximum follow-up period of 45 years. The HR for asthma was stable during the follow-up period. Only the maximum and first five-year follow-ups are reported. In the analysis of variance, age was used as a confounder. The statistical analyses were performed using IBM SPSS Statistics 27 release 27.0.0.0 for Microsoft Windows.

## 3. Results

During the average follow-up of 20.7 years, 14.0% of women and 9.3% of men developed incident asthma. Selected characteristics of the study population at baseline are presented in Table 1.

In both sexes, the risk of asthma increased with increasing BMI. The hazards ratios adjusted for age, area, smoking, physical activity, and study year were 1.00, 1.31 and 1.57 (*p* for the trend <0.001) among women and 1.000, 1.251 and 1.625 (*p* for the trend <0.001) among men (Table 2). Among the other risk factors, smoking was associated with an increased risk of asthma among men (hazard ratio of current smokers versus lifetime non-smokers: 1.137, *p* = 0.004), and physical activity was associated with a decreased risk of asthma among men (hazard ratio active vs. sedentary: 0.938, *p* = 0.032). Risk of asthma attributable to overweight and obesity was 30.8% in women and 19.2% among men. The increase in risk of incident asthma was graded with increasing BMI, and no sudden increase in risk was observed at traditional classification limits of 25 kg/m^2^ or 30 kg/m^2^ (Figure 2).

The association between BMI and the risk of asthma was similar when the follow-up time of the study cohorts was restricted to the first five years (Table 3). The associations of smoking, physical activity and height with asthma risk disappeared when the follow-up time was truncated to five years only. The HR remained stable during the follow-up years.

## 4. Discussion

The results of the present large population-based prospective study demonstrate that overweight and obesity markedly increase the risk of adult-onset asthma among middle-aged people at baseline. High BMI at baseline predicted the future risk of asthma equally in men and women, although the overall risk of asthma was somewhat higher in women. The association between BMI and asthma risk is graded and independent of age, smoking and leisure-time physical activity. Asthma risk increased gradually with BMI, without any threshold limit. In this population, approximately one-fourth of the total asthma incidence was attributed to overweight and obesity (BMI > 25 kg/m^2^). While the highest BMI class failed to reach statistical significance, defined as *p* < 0.05, in the 5-year follow-up, most likely due to the small sample size, the effect size and direction were consistent with maximum follow-up.

Our results are consistent with the findings of the Nurses’ Health Study and the Canadian National Population Health Survey about the strong direct association between BMI and asthma among women [11,12]. In the Nurses’ Health Study II, a direct trend was found across six BMI categories, and female nurses with a BMI of 30 kg/m^2^ or higher had a 2.7-fold increased risk of asthma compared with nurses with a BMI of 20–22.4 kg/m^2^. In the Canadian study, women whose baseline BMI was at least 30 kg/m^2^ had a 1.9-fold increased risk of asthma compared with women whose BMI was 20–24.9 kg/m^2^. In men, the corresponding odds ratio was only 1.1, and it did not significantly differ from unity. Also, a nested case–control study of the participants in the Tucson Epidemiologic Study of Airways Obstructive Disease and the French E3N Cohort Study found an association between obesity and asthma among women only [15,16]. Body fat distribution differs between the sexes, which could explain differences in the BMI–asthma relationship in men and women. However, in a meta-analysis including the French E3N Cohort study, the odds of incident asthma in overweight and obese men and women were found to be similar, indicating that female sex does not modify the obesity–asthma relationship [13]. While female sex was found to be an independent risk factor for incident asthma in another prospective Finnish study of 2620 people aged 24–39, BMI was found to be associated with incidence independent of sex [17].

The definition of asthma diagnosis in large epidemiological studies is challenging. In clinical practice, asthma diagnoses require an objective measure of reversible airway obstruction or, in unclear cases, observation of symptoms and measurement of lung functions over a longer period [10]. Usually, neither of these methods is possible in large population-based studies. The definition of incident cases causes an additional problem because the diagnostic measures should be performed at baseline and repeated during the follow-up.

In the Nurses’ Health Study II and the Canadian National Population Health Survey, baseline data collection was conducted by self-reporting, in addition to the use of case findings collected during the follow-up relayed by self-reporting of asthma diagnoses using repeated postal surveys. The study cohorts were followed four and two years, respectively. Even though the follow-up time was short, both of the studies found a fairly high cumulative asthma incidence. In the Nurses’ Health Study, 1.9 percent developed asthma during the follow-up, while in the Canadian National Population Health Survey, 2.9 percent of women and 1.6 percent of men developed asthma during the follow-up. This may indicate that the participation in the surveys somehow activated the people with incipient asthma to search for medical services, increasing the likelihood of asthma diagnosis. In the Nurses’ Health Study II, the relatively high incidence may also be contributed by the fact that the study population was a selected group of educated people with good access to medical care.

Sin et al. reported an association between obesity and self-reported asthma and found obese subjects to be at lower risk of objective airflow obstruction [18]. In addition to self-reporting, we used two different nationwide register databases in order to exclude people with prevalent asthma at baseline. Therefore, we can assume that the study participants were free of asthma at baseline. During the follow-up, asthma cases were identified using data from the National Social Insurance Institution’s Drug Register. The major advantage of this method is that the follow-up is virtually complete, which minimizes the risk of selection bias in case findings. However, our data do not allow for phenotyping of the patients, which limits the analysis. Many longitudinal studies on asthma and BMI rely on self-reported height and weight, leading to potential biases. In our study, BMI was based on measurements performed by a healthcare professional, which increases the quality of the data.

The etiology of adult-onset asthma is complex. In fact, the association between obesity and the risk of asthma is stronger than that observed with most other putative risk factors. There are several mechanisms that could convey the effect of obesity on pulmonary function. First, obesity and weight gain affect airways mechanically and may cause a decline in pulmonary function [8,19,20]. The most characteristic alteration in pulmonary function found in obesity is a reduction in functional residual capacity (FRC) [6]. Low FRC may increase airway resistance [7]. Furthermore, a reduction in FRC may contribute to the narrowing of small airways and increased airway responsiveness [21,22]. In addition, it has been suggested that a small fraction of adipose tissue in airway walls may be sufficient to produce a clinically meaningful reduction in the cross-sectional area in the airways [23]. Obese individuals are also at increased risk of gastroesophageal reflux disease, which is a known risk factor for adult-onset asthma [24,25]. It has also been shown that weight loss reduces airway obstruction among obese asthma patients [26]. Secondly, obesity may increase the risk of asthma through its influence on progesterone and estrogen levels. Estrogen may be a contributing factor to the excess asthma prevalence in women compared to men during the reproductive years, and estrogen use appears to increase the risk of asthma among postmenopausal women [27,28]. However, the inter-relations between obesity, sex hormones and asthma are complex. Thirdly, airway inflammation is a characteristic feature of bronchial asthma [29]. We have also reported that low-level systemic inflammation is associated with prevalent asthma [30]. Because adipose tissue is a major producer of inflammatory factors [31], obesity may affect airways through its influence on systemic inflammation. Elevated blood triglycerides have been found to be associated with asthma among people with obesity [32]. It is also possible that obesity does not have a causal association with the pathogenesis of asthma but that the people with obesity, for various reasons, develop symptoms more easily, search for medical advice earlier, and receive drug treatment for asthma more frequently than the lean people. Gonzalez-Barcala et al. did not find any association between BMI and hospital admissions due to asthma and concluded that obesity is not a risk factor for exacerbations of asthma [33].

Both obesity and asthma are common and rapidly increasing health problems, particularly in industrialized countries but also in many low- and middle-income countries. In many countries, more than half of the adult population is overweight (BMI over 25 kg/m^2^), 5 to 10 percent have asthma, and a substantial proportion of healthcare cost is used for the treatment of these two disorders [34,35]. Regardless of the exact mechanism of how adiposity might increase asthma risk, the public health implications of the observed relationship between obesity and asthma are potentially large. Even in a situation in which obesity is not directly involved with the pathogenesis of asthma but only accelerates its symptoms, control of obesity could markedly decrease the public health and economic burden caused by asthma.

There are several limitations of the present study. As BMI and other variables used in the analysis were measured only once at baseline and the follow-up time was up to 45 years, significant changes may have occurred between the measurements and the diagnosis. However, in the sensitivity analysis, there was only insignificant variation in the estimates when the follow-up was truncated to 5 years. We also relied on register data for asthma diagnosis. As reimbursement rights in Finland are based on objective clinical findings and set criteria, the cases are most probably clinically significant asthma with objective airway obstruction; on the other hand, mild asthma cases may not have been ascertained.

We only included BMI and no other adiposity indicators in our analyses. Several studies have shown that other adiposity indicators, such as waist circumference, waist–hip ratio, and waist–height ratio, are also associated with the risk of asthma [36,37,38,39]. We have data on waist and hip circumference for our study cohorts since the 1987 survey, and in the future, we will be able to evaluate the importance of these other adiposity indicators for the risk of asthma. We also did not include dietary factors, which could have provided additional insights into the relationship between BMI and asthma incidence.

## 5. Conclusions

In conclusion, obesity and overweight markedly increase the risk of adult-onset asthma among middle-aged men and women. No sex difference was found in terms of this risk. The increase in prevalence of obesity may be a contributing factor to the increase in asthma prevalence observed in many populations.

## Figures and Tables

**Figure 1 nutrients-16-02515-f001:**
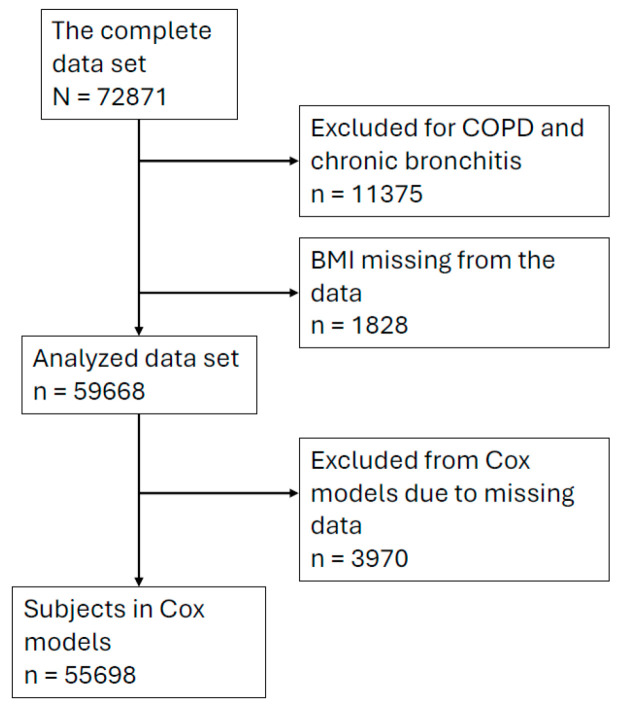
The selection of participants for the data set used in the final analysis.

**Figure 2 nutrients-16-02515-f002:**
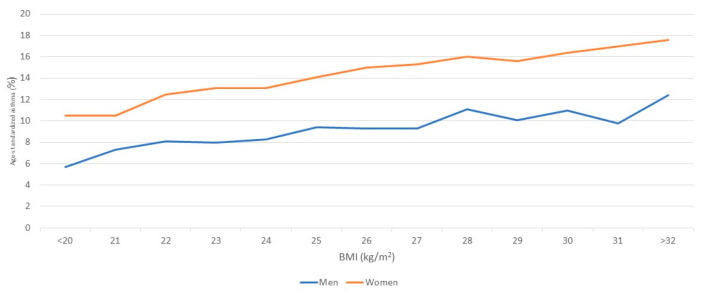
Age-standardized asthma as a function of BMI for men and women.

**Table 1 nutrients-16-02515-t001:** Selected characteristics of study participants at baseline according to outcome, i.e., incident adult-onset asthma during follow-up. * *t*-test for continuous variables and chi-square test for categorical variables.

	Women			Men		
	No Asthma	Asthma	*p*-Value *	No Asthma	Asthma	*p*-Value *
Number	27,901	4548		24,680	2539	
Age (years)	45.8	45.2	0.002	45.6	45.2	0.062
Body mass index (kg/m^2^)	26.1	26.7	<0.001	26.5	27.1	<0.001
Height (cm)	161.1	160.8	0.004	174.6	174.5	0.701
Body mass index class			<0.001			<0.001
Below 25 (%)	48.0	42.1		37.2	31.9	
25–29.9 (%)	33.2	35.1		46.7	48.4	
30 or over (%)	18.8	22.8		16.1	19.7	
Smoking			0.011			<0.001
Current smoker (%)	14.2	14.3		30.5	25.2	
Ex-smoker (%)	11.8	13.3		28.6	33.8	
Never smoker (%)	74.0	72.4		40.9	41.0	
Physical activity Sedentary (%)	32.1	33.6	0.063	24.3	26.2	0.088
Moderate (%)	51.0	50.5		51.5	50.6	
Active (%)	17.0	15.9		24.2	23.1	

**Table 2 nutrients-16-02515-t002:** Cox model for incident asthma during the maximum follow-up (up to 45 years).

	Women (N = 30,099)				Men (N = 25,599)			
	HR	*p*	CI	n	HR	*p*	CI	n
Age (years)	0.993	<0.001	0.990–0.996		1.000	0.830	0.996–1.005	
Body mass index class								
Below 25 kg/m^2^	Ref	Ref	Ref	14,490	Ref	Ref	Ref	9512
25–29.9 kg/m^2^	1.320	<0.001	1.220–1.428	10,042	1.265	<0.001	1.143–1.401	11,960
30 or over kg/m^2^	1.531	<0.001	1.389–1.688	5567	1.618	<0.001	1.412–1.853	4127
Smoking								
Never smoker	Ref	Ref	Ref	22,162	Ref	Ref	Ref	10,479
Ex-smoker	1.146	0.012	1.030–1.275	3620	1.153	0.009	1.036–1.283	7417
Current smoker	1.099	0.059	0.996–1.212	4317	0.981	0.738	0.879–1.096	7703
Educational tertile	0.976	0.251	0.936–1.017		0.976	0.412	0.922–1.034	
Physical activity	0.985	0.574	0.936–1.037		0.942	0.072	0.882–1.005	

**Table 3 nutrients-16-02515-t003:** Cox model for incident asthma during the first five years of follow-up.

	Women (N = 935)				Men (N = 1106)			
	HR	*p*	CI	n	HR	*p*	CI	n
Age (years)	0.979	<0.001	0.973–0.985		0.962	<0.001	0.953–0.971	
Body mass index class								
Below 25 kg/m^2^	Ref	Ref	Ref	356	Ref	Ref	Ref	321
25–29.9 kg/m^2^	1.281	0.017	1.046–1.569	329	1.433	0.019	1.060–1.936	527
30 kg/m^2^ or over	1.258	0.062	0.988–1.602	250	1.357	0.086	0.958–1.921	258
Smoking								
Never smoker	Ref	Ref	Ref	650	Ref	Ref	Ref	316
Ex-smoker	1.258	0.064	0.988–1.602	132	1.1070	0.630	0.813–1.408	362
Current smoker	0.930	0.561	0.729–1.1187	153	0.567	<0.001	0.411–0.782	428
Education tertile	1.053	0.372	0.940–1.178		1.016	0.834	0.876–1.179	
Physical activity	1.135	0.048	1.001–1.287		1.050	0.598	0.875–1.260	

## Data Availability

The datasets presented in this article are not readily available due to national legislation on medical research. Requests to access the datasets should be directed to Finnish Institute for Health and Welfare.

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
