# Peer review of "Body Mass Index and the Risk of Adult-Onset Asthma: A Prospective Observational Study among 59,668 Middle-Aged Men and Women in Finland"

_nutrients, 2024, doi:10.3390/nu16152515_

Round 1
Reviewer 1 Report
Comments and Suggestions for Authors
This manuscript presents findings from a large population-based prospective study conducted in Finland, aiming to investigate the correlation between overweight and obesity and the incidence of adult-onset asthma. Notably, both asthma diagnosis and BMI measurements were conducted by healthcare professionals, ensuring data quality. The analysis revealed that a high BMI at baseline predicted future asthma risk in both men and women, although women showed a slightly higher overall asthma risk. Importantly, this association between BMI and asthma risk was graded and independent of factors such as age, smoking, and leisure time physical activity. These results contribute valuable additional evidence supporting the notion that obesity and overweight significantly elevate the risk of adult-onset asthma.
Major comments:
1. The discrepancy in the total number of participants raises concerns. While the authors initially claim that a total of 60,639 Finnish men and women participated in the study, Table 2 presents different figures as a total of 64126 (N=33,867 for women and N=30,259 for men). This inconsistency prompts the need for a re-examination of the analysis to ensure data accuracy and reliability.
2. The definition of "age-standardized asthma" and the method of its calculation in Figure 1 are not clearly outlined. Additionally, for enhanced clarity and informativeness, it would be beneficial to break down the correlations presented in Figure 1 into separate figures (e.g., A, B, and C) either by age range or by BMI range.
3. The baseline data presented in Table 1 appears to be ambiguous. If the number of asthma cases present those adult-onset asthma, clarification is needed regarding when the BMI was measured—was it at baseline or at the time of asthma diagnosis? Additionally, details on how the p-value was calculated and whether it was consistent across different BMI classes are necessary for understanding the statistical significance of the findings.
4. The novelty of this study could be enhanced. Several epidemiological studies have already demonstrated a significant increase in the risk of adult-onset asthma associated with obesity and overweight. While it is indeed valuable to reaffirm these conclusions with a different population in a cohort study, the novelty could be further heightened by investigating the correlation of obesity-related parameters, such as hypertension, diabetes, hyperlipidemia, etc., with late-onset asthma within the study's database.
5. The race information should be included in both Table 1 and 2 since race is another factor may affect the correlation between obesity and asthma risk. Given that the fact that Finns comprise 89.8% of the Finland population, this aspect renders the analysis and conclusions of this study unique and crucial. Comparing data from this study with that from other populations will yield more valuable insights.
Minor comments:
1. The introduction should include the definition of adult-onset asthma and provide the rationale for focusing on adult-onset asthma for this study.
2. A figure legend needs to be included for Figure 1.
3. The resolution of Figure 1 is insufficient.
4. Although education was included as a factor in data analysis, it was not discussed.
5. It would be clearer to present the confidence interval (CI) data in Table 3 in one row, as done in Tables 1 and 2.
6. What is the p-value calculated for age, and what is the significance of comparing ages?
Author Response
Reviewer 1:
This manuscript presents findings from a large population-based prospective study conducted in Finland, aiming to investigate the correlation between overweight and obesity and the incidence of adult-onset asthma. Notably, both asthma diagnosis and BMI measurements were conducted by healthcare professionals, ensuring data quality. The analysis revealed that a high BMI at baseline predicted future asthma risk in both men and women, although women showed a slightly higher overall asthma risk. Importantly, this association between BMI and asthma risk was graded and independent of factors such as age, smoking, and leisure time physical activity. These results contribute valuable additional evidence supporting the notion that obesity and overweight significantly elevate the risk of adult-onset asthma.
Major comments:
- The discrepancy in the total number of participants raises concerns. While the authors initially claim that a total of 60,639 Finnish men and women participated in the study, Table 2 presents different figures as a total of 64126 (N=33,867 for women and N=30,259 for men). This inconsistency prompts the need for a re-examination of the analysis to ensure data accuracy and reliability.
The Cox-model in the original manuscript included people who had chronic bronchitis. This has now been amended. The changes were otherwise insignificant, but in 5 year follow up the highest BMI class the p-value increased above the traditional 0.05 threshold most likely due to smaller sample size. However, the effect size and direction are consistent with the 45-year model and this has now been addressed in the discussion section.
- The definition of "age-standardized asthma" and the method of its calculation in Figure 1 are not clearly outlined. Additionally, for enhanced clarity and informativeness, it would be beneficial to break down the correlations presented in Figure 1 into separate figures (e.g., A, B, and C) either by age range or by BMI range.
Age was used as covariate in analysis of variance. The effect of age was small and breaking down the figure would not add any relevant information. The goal of the figure is to show that the BMI associated risk of asthma has graded trend and the BMI category boundaries are arbitrary in this context. Following has been added to the methods section: “In analysis of variance age was used as a cofactor.”
- The baseline data presented in Table 1 appears to be ambiguous. If the number of asthma cases present those adult-onset asthma, clarification is needed regarding when the BMI was measured—was it at baseline or at the time of asthma diagnosis? Additionally, details on how the p-value was calculated and whether it was consistent across different BMI classes are necessary for understanding the statistical significance of the findings.
The BMI (along with all other factors) was measured only at baseline and the follow up was conducted only through registers. This has now been emphasized in the Methods section and has already been stated in limitations paragraph of the Discussion. The caption has been amended to avoid any confusions and the text now reads “Selected characteristics of study participants at baseline by the outcome i.e. incident adult on-set asthma during the follow-up”. For p-values chi2 was used to detect differences in the distributions as usual.
- The novelty of this study could be enhanced. Several epidemiological studies have already demonstrated a significant increase in the risk of adult-onset asthma associated with obesity and overweight. While it is indeed valuable to reaffirm these conclusions with a different population in a cohort study, the novelty could be further heightened by investigating the correlation of obesity-related parameters, such as hypertension, diabetes, hyperlipidemia, etc., with late-onset asthma within the study's database.
While including other issues proposed would enhance the novelty value of this work it is, unfortunately, out of the scope of this manuscript. However, we believe that confirming the obesity associated risk on adult on set asthma in a large population samples already has a marked scientific significance.
- The race information should be included in both Table 1 and 2 since race is another factor may affect the correlation between obesity and asthma risk. Given that the fact that Finns comprise 89.8% of the Finland population, this aspect renders the analysis and conclusions of this study unique and crucial. Comparing data from this study with that from other populations will yield more valuable insights.
In these data sets there are very few people whose ethnicity is not Finnish and the questionnaires have originally (in the 1970s) not included any questions on race. This has been amended in the future iterations of the study.
Minor comments:
- The introduction should include the definition of adult-onset asthma and provide the rationale for focusing on adult-onset asthma for this study.
We have now included the following definition from GINA 2024 in the first paragraph of the Introduction. This now gives the background and rationale when the aims of the study are presented in the final paragraph of the Introduction.
“Adult-onset asthma is defined as asthma presenting for first time in adults life. It tends to be non-allergic and often does not response to inhaled corticosteroids as well as allergic asthma. Similarly, asthma with obesity may present with different patterns of airway inflammation, but typically with limited eosinophilic inflammation.”
- A figure legend needs to be included for Figure 1.
Figure 1 legend is included in the manuscript. Unfortunately, the figure was in the end of p. 11 so the legend was in the beginning of p. 12.
- The resolution of Figure 1 is insufficient.
We have now included the figure in vector format.
- Although education was included as a factor in data analysis, it was not discussed.
Although the aim of this study was not related to education, we felt important to control for it as a confounding factor as lower education is typically associated with higher BMI. The education does not seem to be an important factor in this context. Therefore, we have focused the discussion on BMI to present the results in clear and concise manner.
- It would be clearer to present the confidence interval (CI) data in Table 3 in one row, as done in Tables 1 and 2.
Table 2 and 3 are identical in formatting. In the table describing demographics (table 1) Cis are not typically included.
- What is the p-value calculated for age, and what is the significance of comparing ages?
From the question it is not evident whether the reviewer refers to demographics in table 1 or Cox models in tables 2 and 3. The p-values are reported in all of these tables. In demographics we feel that it is important to report any differences between the groups (table 1) as it allows readers to assess the validity of the comparison. In the Cox-models we wanted to report it explicitly as the effect was not consistent across all the models.
Reviewer 2 Report
Comments and Suggestions for Authors
I appreciate the opportunity to review the manuscript for publication in MDPI nutrients. The manuscript provides valuable insights into the characteristics and risk factors associated with asthma etiology. The study includes a large sample size, long follow-up period, and detailed methodological approach. I feel that the topics are interesting. I have a few comments as follows.
BMI Calculation: The rationale for choosing the specific BMI categories (<24.9, 25-29.9, >30 kg/m²) should be explicitly stated with relevant reference citation.
Methods:
Further information on the accuracy and completeness of the registers including the process for linking data through national registers is required to strengthen the credibility of the follow-up data.
Additional figure of a flowchart depicting the selection process and exclusions is necessary to enhance transparency.
Table 1: The alignment and spacing should be amended for better reader-friendly fashions.
Discussion:
The authors had better provide a more detailed exploration of the potential mechanisms underlying the observed association between high BMI and increased asthma risk. While the section mentions that body fat distribution differs between genders, it does not delve into how these differences might physiologically contribute to asthma risk. More detailed hypotheses or pathological mechanisms regarding gender-specific biological function would be necessary.
Some of the references are published more than decades ago and could be updated to include the most recent studies in the BA etiology field.
Author Response
I appreciate the opportunity to review the manuscript for publication in MDPI nutrients. The manuscript provides valuable insights into the characteristics and risk factors associated with asthma etiology. The study includes a large sample size, long follow-up period, and detailed methodological approach. I feel that the topics are interesting. I have a few comments as follows.
BMI Calculation: The rationale for choosing the specific BMI categories (<24.9, 25-29.9, >30 kg/m²) should be explicitly stated with relevant reference citation.
The categories stem from the established limits defined by WHO and CDC. They define the upper limit of healthy weight as 24.9, upper limit of overweight as 29.9 and obesity as > 30. The limits are commonly acknowledged in both research and clinical communities.
Methods:
Further information on the accuracy and completeness of the registers including the process for linking data through national registers is required to strengthen the credibility of the follow-up data.
The registers in Finland are reliable and complete. All residents in Finland have personal identifier which can be used throughout the registers. The registers used are centralized and nation wide and they have responsibility to share the data for research use.
Additional figure of a flowchart depicting the selection process and exclusions is necessary to enhance transparency.
We have added a new figure to describe the effects of exclusions and missing data.
Table 1: The alignment and spacing should be amended for better reader-friendly fashions.
We trust that the editorial team will amend this when producing the final lay out for printing if the manuscript is accepted.
Discussion:
The authors had better provide a more detailed exploration of the potential mechanisms underlying the observed association between high BMI and increased asthma risk. While the section mentions that body fat distribution differs between genders, it does not delve into how these differences might physiologically contribute to asthma risk. More detailed hypotheses or pathological mechanisms regarding gender-specific biological function would be necessary.
Several proposed mechanisms have been discussed in the sixth paragraph of the discussion and a new reference discussing role of adipose tissue in airway walls has been added. However, more detailed mechanistic discussion on cellular or physiological level is outside the scope of this epidemiological study.
Some of the references are published more than decades ago and could be updated to include the most recent studies in the BA etiology field.
As there is more literature on the subject that can be included in any single manuscript we prefer to cite the original publications rather than confirmatory papers to credit the original authors for the conceptualization of the phenomena. We have cited the most recent papers in cases where there has been conceptual advances either in the results or methodology.